# First isolation of viable *Toxoplasma gondii* from a black mangabey (*Lophocebus aterrimus*) reveals the emergence of the *Africa* 1 lineage in East Asia

**Yiheng Ma, Liulu Yang, Yurong Yang***

**College of Veterinary Medicine, Henan Agricultural University**, Zhengzhou, China

* yangyu7712@sina.com, yryang@henau.edu.cn

## Abstract

*Toxoplasma gondii* is an intracellular protozoan whose intermediate hosts encompass nearly all warm-blooded animals, including humans and non-human primates (NHPs). In this study, molecular, serological, immunohistochemical and bioassay methods were used to investigate *T. gondii* infection in 17 captived NHPs in zoos from China between 2022 and 2023. The infection rate was higher in New World NHPs (83.3%) than in Old World NHPs (16.7%) (*P* = 0.2424). A viable strain of *T. gondii* was successfully isolated from the tissues of a black mangabey (*Lophocebus aterrimus*). The strain designated as TgMonkeyCHn3 was genotyped as ToxoDB #6, which had demonstrated virulence in Swiss Webster outbred mice. The ROP18/ROP5 allele combination was identified as 1/3. The emergence of this strain highlights the increased genetic diversity of *T. gondii* in East Asia and presents new challenges for the prevention of toxoplasmosis, particularly in captive NHPs. To the best of our knowledge, this is the first report of the ToxoDB genotype #6 isolated in East Asia and the black mangabey is a new host for *T. gondii*.

## Author summary

*Toxoplasma gondii* is a globally distributed intracellular parasite with marked genetic diversity, especially in South America. China has long been considered a region with a limited diversity of *T. gondii*, being dominated by the *Chinese* 1 lineage (ToxoDB #9). This study reports the first identification and characterization of the *Africa* 1 strain (ToxoDB #6) in China isolated from a captive black mangabey native to tropical Africa. The emergence of *Africa* 1 augments the genetic diversity of *T. gondii* in East Asia and likely stems from the importation of African wildlife. The appearance of the strain emphasizes the role of human activities in the transcontinental transmission and geographical distribution of zoonotic pathogens. Overall, the isolated strain of non-native genotypes like Africa 1 tend to exhibit high virulence, which may complicate local toxoplasmosis control. To

**Data availability statement:** All relevant data are within the manuscript and its Supporting Information files.

**Funding:** This work was supported by the Henan Province's International Scientific and Technological Cooperation Projects (reference 242102521041 to YY). The funders had no role in study design, data collection and analysis, decision to publish, or preparation of the manuscript.

**Competing interests:** The authors have declared that no competing interests exist.

mitigate zoonotic risks, more infection investigations and stricter animal entry quarantines are needed to monitor emerging *T. gondii* genotypes and address new challenges in controlling the pathogen.

## 1. Introduction

*Toxoplasma gondii* is an intracellular protozoan whose intermediate hosts span virtually all warm-blooded animals, including humans and non-human primates (NHPs). Felines are definitive hosts and can shed oocysts in their feces, thereby playing a critical role in the transmission of *T. gondii* [1]. The pathogenicity of *T. gondii* is related to the virulence of the strain and the host species [2].

NHPs serve as a critical translational bridge between fundamental research and clinical applications due to their high degree of genetic homology with humans [3,4]. Understanding the pathogenicity of *T. gondii* in NHPs may thus provide valuable insights into toxoplasmosis in humans. Importantly, the behavior of *T. gondii* exhibits host-specific characteristics [5]. New World NHPs show high susceptibility to *T. gondii* and tend to develop severe toxoplasmosis, often dying without apparent clinical signs or with nonspecific symptoms such as anorexia and depression [1,6]. In contrast, Old World NHPs are considered resistant to *T. gondii* [7]. The reasons for this difference are unclear, but host-parasite co-evolution is thought to play an important role. Most New World NHPs may be minimally exposed to *T. gondii* oocysts in soil and feline feces due to their arboreal lifestyle, whereas most Old World NHPs stay on the ground during foraging, an activity that increases the risk of exposure to *T. gondii*. As a result, Old World NHPs have acquired greater natural resistance under the evolutionary pressure of natural selection [7,8].

Virulence of *T. gondii* strains in hosts varies with the genetic background of the parasites [9]. To date, the *T. gondii* genotypes identified in NHPs worldwide include ToxoDB genotypes #1, #3, #4, #9, #11, #13, #21, #36, #163, and #206 and seven mixed genotypes, indicating high genetic diversity of *T. gondii* strains in NHPs [6,10–18]. In China, only limited ToxoDB genotypes in NHPs have been identified: ToxoDB genotype #9 from *Cebus apella* and *Erythrocebus patas*, and mixed genotypes from *Macaca mulatta*, *Saimiri vanzolinii*, and *Lemur catta* [10,11,16–18]. Among these, ToxoDB #9 (*Chinese* I) is the predominant genotype in China; however, the emergence of new genotypes presents additional challenges for the conservation of endangered NHP species.

The objective of the present study is to investigate *T. gondii* infection in captive NHPs and explore the genotype, pathogenicity, and epidemiological origin of *T. gondii* isolate.

## 2. Materials and methods

### 2.1. Ethics statement

The protocol was approved by the Beijing Association for Science and Technology (SYXK [Beijing] 2007–0023). All animals were handled following the Animal Ethics

Procedures and Guidelines of China. All experiments reported here were approved by the Institutional Animal Use Protocol Committee of Henan Agricultural University, China.

Verbal consent was obtained to collect samples from zoos. This method can be used in China, and it was approved by the Ethics Committee of Henan Agricultural University.

## 2.2. NHP samples collection and necropsy

Between January 2022 and December 2023, 16 NHPs died in zoos located in Shandong, Henan, and Guangdong provinces of China (S1 Table). The workflow is shown in S1 Fig. Tissues, including liver, tongue, spleen, kidneys, heart, lungs, brain, mesenteric lymph nodes, pancreas, and skeletal muscle, were collected from 16 cases (5 fresh and 11 formalin-fixed). Separate sterile bags were used for each fresh tissue during autopsy sampling to avoid cross-contamination. Additionally, serum samples were obtained from a healthy Bornean orangutan (*Pongo pygmaeus*) (S1 Table). These samples were sent to the Laboratory of Veterinary Pathology, Henan Agricultural University (Zhengzhou, Henan, China) for pathology evaluation, survey for etiology, and tests for *T. gondii* infection.

## 2.3. Detection of *T. gondii* antibodies by a modified agglutination test (MAT)

NHP samples were tested for *T. gondii* antibody by MAT [9,19]. The whole formalin-treated tachyzoites of the *T. gondii* RH strain were obtained from the University of Tennessee (Knoxville, TN, USA). All heart juice or serum samples were tested using a 1:2 ratio and two-fold serial dilution to the final titers. Samples with titers of 1:8 or higher were considered positive for *T. gondii* [10,11]. Positive and negative controls were included in each test.

## 2.4. PCR identification of *T. gondii* in tissues of NHPs

Genomic DNA was extracted from 30 mg of fresh tissue (liver, tongue, spleen, kidney, heart, lungs, brain, mesenteric lymph nodes, pancreas, and skeletal muscle) using a DP304 DNA extraction kit (Tiangen Biotech, Beijing, China) and eluted into 100 µL of Tris-EDTA buffer. *T. gondii* was detected using the primer pair TOX-5 (5′-CGC TGC AGA CAC AGT GCA TCT GGA TT-3′) and TOX-8 (5′-CCC AGC TGC GTC TGT CGG GAT-3′) [20]. The reaction system (25 µL) contained 12.5 µL of 2 × PCR Mix (GDSBio, Guangzhou, China), with a concentration of 0.2 µM for each set of primers. Target DNA was amplified using the following program: initial denaturation at 94°C for 5 min, then denaturation at 94°C for 1 min, annealing at 60°C for 1 min, and extension at 72°C for 1 min, following 35 cycles, an extension step of 10 min at 72°C was added. The expected length of the target DNA was 450 bp. All PCRs were run with negative and positive controls (*T. gondii* RH strain DNA).

## 2.5. Histopathology and immunohistochemistry

After fixation with a 10% neutral-buffered formalin solution, all tissue samples (S1 Table) from NHPs were processed using routine histological techniques and stained with hematoxylin and eosin (H&E). Immunohistochemical (IHC) staining was also performed on the tissues of NHPs (n = 16). The polyclonal rabbit anti-*T. gondii* antibody was provided by Dr. J. P. Dubey (Beltsville, Agricultural Research Service, USDA, MD, USA). The anti-rabbit IgG conjugated with HRP/DAB IHC detection kit (ab64264) was purchased from Abcam (Cambridge, UK). Brain tissue sections of mice infected with *T. gondii* VEG strain (provided by Dr. Dubey) were used as positive controls for IHC staining. Briefly, IHC staining was performed on paraffin-embedded tissue sections following deparaffinization and rehydration. Antigen retrieval was conducted via microwave heating in citrate buffer (pH 6.0). Endogenous peroxidase activity was blocked with 3% $H_2O_2$. The sections were incubated overnight at 4°C with primary antibody at a dilution of 1:3,000. The secondary antibody was applied for 15 min at 37°C. Signals were amplified with streptavidin-peroxidase and then visualized with DAB under a microscope. The sections were counterstained with hematoxylin, dehydrated, and mounted in neutral balsam.

## 2.6. Isolation of *T. gondii* strains from NHPs by mouse bioassays

Tissue samples (heart, tongue, brain, and leg muscle) from five NHPs were bioassay in mice. Briefly, the tissue homogenate (50 g) was digested with pepsin (5.2 g hog stomach pepsin, 10.0 g NaCl, and 14 mL HCl, with water added to a volume of 1000 mL, pH 1.10–1.20) at 37°C for 60 min with shaking [1]. Subsequent filtration through gauze and centrifugation separated the sediment, which was neutralized using sodium bicarbonate (1.2%, pH 8.3) with a phenol red indicator (producing a color change to orange). IFN-γ knockout mice were used for bioassay, as they lacked Th1 immune responses, leading to parasite replication and the isolation of large numbers of tachyzoites [1]. The pepsin-digested products (1 mL per mouse) were subcutaneously injected into *T. gondii*-free Swiss Webster outbred mice (n = 4–5) and/or IFN-γ knockout mice (n = 1–2).

Each group was sequentially named Tox#20-x. For example, in "Tox#20-50", "Tox" represents *T. gondii* bioassay; "20" represents the species ID of the NHPs, and "50" represents the group ID. Mice were euthanized when they showed severe signs: respiratory distress, unwillingness to move or eat, neurological symptoms, or hypothermia. Tissues (lungs, mesenteric lymph nodes, or brain) of dead mice were examined microscopically for the presence of *T. gondii* parasites via smears [1]. At 30 days of post-infection (DPI), blood samples were collected via facial vein puncture from surviving mice and tested for *T. gondii* antibodies by MAT.

## 2.7. *In vitro* cultivation and genotyping of *T. gondii*

CV-1 cells were routinely cultured in Roswell Park Memorial Institute (RPMI) 1640 medium supplemented with 3% fetal bovine serum (FBS) at 37°C in a 5% $CO_2$ humidified incubator. *T. gondii*-positive lung, mesenteric lymph node, and brain tissues examined by smears from euthanized mice were homogenized in sterile phosphate-buffered saline (PBS) and then seeded onto CV-1 cells [1]. After 1–2 h of incubation, the mouse tissues were removed, and fresh RPMI 1640 medium was added to the flask. The growth status of tachyzoites was observed every other day, and tachyzoites were collected when 80% of CV-1 cells ruptured to release tachyzoites.

DNA was extracted from cell culture tachyzoites, and the *T. gondii* genotype was identified using multilocus nested PCR-RFLP typing with 10 genetic markers (SAG1, SAG2, SAG3, BTUB, GRA6, c22-8, c29-2, L358, PK1, and Apico) as previously described by Su et al. [21]. The virulence genes ROP18 and ROP5 were also typed according to previous studies [22]. Reference *T. gondii* strains' (GT1, PTG, CTG, MAS, TgCgCa1, TgCatBr5, TgCatBr64, and TgRsCr1) DNA was kindly provided by Dr. Chunlei Su (University of Tennessee, Knoxville, TN, USA). The negative control was DNA-free water.

## 2.8. Transmission electron microscopy (TEM)

The sample preparation procedure for examining *T. gondii* tachyzoites under TEM was as follows: when there were many parasitophorous vacuoles, a 2.5% glutaral-paraformaldehyde mixture was added to the cell culture flask for pre-fixation, after which the cells were removed with a cell scraper and centrifuged at 500 × g for 10 min, and the supernatant was discarded. A glutaral-paraformaldehyde mixture was then added to the precipitate for fixation, and the samples were sent to Henan University of Chinese Medicine for TEM section production.

## 2.9. Virulence evaluation

The virulence of *T. gondii* isolated from NHP tissues was evaluated in *T. gondii*-free Swiss Webster outbred mice [9]. The concentration of tachyzoites collected from cell cultures was determined using a hemocytometer. Subsequently, 10-fold serial dilution concentrations from $1 \times 10^4$ to <1 tachyzoite/mL were prepared using sterile PBS. Each group of Swiss mice (n = 5) was inoculated by intraperitoneal injection with 1 mL of tachyzoites at different dilutions. Severely ill mice were euthanized and examined microscopically for *T. gondii* by lung and brain smears. At 30 DPI, blood samples were collected

from the surviving mice and tested for *T. gondii* antibodies by MAT. All clinical symptoms were recorded daily up to 60 DPI. The virulence of the *T. gondii* strain was evaluated according to the percentage of dead mice among the *T. gondii*-positive mice.

## 2.10. Statistical analysis

For categorical variables, the differences in the *T. gondii* infection rate were analyzed using the Chi-square test. The 95% confidence intervals (CIs) for infection rates were calculated to estimate the precision of the statistical results. Data were expressed as the mean ± SEM. Kaplan-Meier survival curve analysis followed by a log-rank (Mantel-Cox) test was used to compare the survival time between the groups or strains. All statistical analyses were performed with GraphPad Prism 8.0 (GraphPad Software Inc., San Diego, CA, USA). *P* values < 0.05 were considered statistically significant.

## 3. Results

### 3.1. Pathological examination

The background, clinical symptoms, pathological changes, and *T. gondii* infection information of NHPs (n = 17) are summarized in S1 Table. Case #43 was alive with no visible abnormality until December 31, 2024. Among the 16 dead cases, interstitial pneumonia (6/16), jaundice (3/16), myocarditis (3/16), aplasia (3/16), suppurative pneumonia (2/16), hypoproteinemia (2/16), malignant tumors (2/16), and fungal infection granulomas (1/16) were identified by gross and histopathological observation.

Of the 16 NHPs with detailed IHC examination data, *T. gondii* tachyzoites were found in the tissues of four New World NHPs (cases #29, #30, #33, and #34) and one ring-tailed lemur (case #41), primarily in the lungs and spleen (S2 Fig). In addition, *T. gondii* tissue cysts were found in the myocardium of the black mangabey (case #35).

### 3.2. *T. gondii* infection by MAT and PCR

*T. gondii* antibody was examined in the serum or heart juice samples from six NHPs by MAT. Fifty percent (3/6) (95% CI, 18.76%–81.24%) of the NHPs showed *T. gondii* antibodies, with titers of 1:32 in a black mangabey (case #35), 1:8,192 in an orangutan (case #43), and 1:1,600 in a black-capped capuchin (case #30). Furthermore, *T. gondii* DNA was examined in tissue samples from five NHPs by PCR and only detected in the heart and lung of the black mangabey (case #35). Based on the combined results of MAT, PCR, IHC, the infection rate of *T. gondii* in New World NHPs was 66.7% (4/6) (95% CI, 29.57%–90.75%), which was higher than that in Old World NHPs at 16.7% (1/6) (95% CI, 11.40%–58.22%); however, the difference was not significant (*P* = 0.2424). In addition, the infection rate of *T. gondii* was 33.3% (1/3) (95% CI, 5.63%–79.76%) in Hominoidea and 50.0% (1/2) (95% CI, 9.45%–90.55%) in Lemuriformes, and there was no significant difference among the four NHP groups (New World NHPs, Old World NHPs, Hominoidea, and Lemuriformes).

### 3.3. Isolation, TEM and genotyping of viable *T. gondii* strains

The tissues of five NHPs were individually bioassayed in mice, and a *T. gondii* strain was successfully isolated from a black mangabey (case #35) (S5 Table). In the remaining four cases that were bioassayed in mice (cases #30, #40, #41, and #44), none of the mice (n = 5) had *T. gondii* antibodies at 60 DPI according to MAT or the presence of tissue cysts in brain smears.

In the group Tox#20–53 (inoculated with pepsin-digested liquid from case #35), M#999 and M#997 were euthanized for acute toxoplasmosis at 18 DPI and 30 DPI, respectively, and *T. gondii* cysts (1330 ± 110) were observed in brain smears (S3A Fig). Tissues of M#999 and M#997 were ground and then inoculated into new groups, Tox#20–54 (two Swiss mice, one IFN-γ $^{-/-}$ mouse) and Tox#20–55 (five Swiss mice), respectively. In the group Tox#20–54, *T. gondii* parasites were observed in the brain and lung of M#7 and M#8 at 15 DPI (S3B Fig). In the group Tox#20–55, four of the five mice had *T.*

*gondii* parasites in their brains (S3C Fig) or lungs. The surviving mice in the above three groups were seropositive for *T. gondii* at 30 DPI.

The *T. gondii* tachyzoites from the lungs of M#22 in group Tox#20–55 were successfully propagated in CV-1 cell cultures (9 DPI) and designated as TgMonkeyCHn3. The findings for *T. gondii* were confirmed by ultrastructural analysis (S3D-S3F Fig). The tachyzoite consisted of an apical ring, a conoid nucleus, several rhoptries ($5.1\pm1.1$), micronemes ($15.1\pm4.3$), lipid bodies ($1.6\pm0.6$), dense granules ($5.8\pm1.3$), and amylopectin ($3.5\pm0.5$). The TgMonkeyCHn3 strain was determined to be ToxoDB genotype #6 (also known as *Africa* 1 or BrI) by PCR-RFLP (S2 Table, S4 and S5 Figs). Furthermore, the virulence gene type of the ROP18/ROP5 allele was 1/3 (S2 Table and S4 and S5 Figs).

### 3.4. Virulence evaluation of TgMonkeyCHn3

There was 71.4% mortality (15/21, CI: 49.79%–86.44%) for TgMonkeyCHn3-infected Swiss mice. TgMonkeyCHn3 ($\geq 10^2$ tachyzoites) caused acute death in infected Swiss mice at $11.4\pm3.7$ DPI, and the survival time decreased with increasing doses of tachyzoite inoculation (Fig 1 and S3 Table). The average survival times for mice inoculated with $10^4$, $10^3$, and $10^2$ tachyzoites were $10.0\pm0.6$, $10.2\pm0.7$, and $14.5\pm3.0$ DPI, respectively. 85.7% (6/7) mice infected with 1 or $10^1$ TgMonkeyCHn3 tachyzoites were confirmed by MAT to be chronically infected with *T. gondii* at 30 DPI. A log-rank test demonstrated an overall significant difference in survival times between all groups of mice ($\chi^2=34.43$, $df=6$, $P<0.0001$). Pairwise comparisons revealed that only mice infected with $10^3$ tachyzoites had a significantly lower survival time in comparison to the 10 tachyzoites group ($P=0.0255$). Acutely infected mice showed typical pathological characteristics of toxoplasmosis (encephalitis, interstitial pneumonia, lymph node enlargement, pleural exudate, and ascites). Tachyzoites or cysts were observed in smears of lungs, peritoneal effusion, and brains.

## 4. Discussion

In this study, the infection rate of *T. gondii* in captive NHPs from Chinese zoos was 41.2% (7/17) (95% CI, 21.56%–64.05%) based on the combined results of MAT, PCR, IHC, and mouse bioassays. The infection rate of *T. gondii* in New World NHPs was higher than that in Old World NHPs ($P=0.2424$). Numerous reports have indicated the high susceptibility of New World NHPs to *T. gondii*, whereas Old World NHPs were considered resistant to *T. gondii* and could survive during chronic infection [6,10–13,23]. Of the six New World NHPs, case #30 (*Sapajus apella*) was seropositive with a titer of 1:1,600. Taking together, the IHC and pathological findings suggest that *T. gondii* infection may be a possible contributing

**Fig 1. Kaplan-Meier survival curve of Swiss mice infected with various doses of *T. gondii* TgMonkeyCHn3 strain tachyzoites.** Log-rank test demonstrated an overall significant difference in survival rate between all groups of mice ($\chi^2=34.43$, $df=6$, $P<0.0001$). Pairwise comparisons revealed that only mice infected with $10^3$ tachyzoites had a significantly lower survival time in comparison to mice infected with 10 tachyzoites ($P=0.0255$).

factor in the deaths of four New World NHPs (cases #29, #30, #33, and #34), which supports that New World NHPs are highly susceptible to *T. gondii.* However, the blood samples and fresh tissues necessary for in-depth study were unfortunately lacking other than case #30.

A viable *T. gondii* strain, TgMonkeyCHn3, was isolated from a black mangabey (case #35), and toxoplasmosis was confirmed by MAT (1:32), PCR (heart and lungs), and IHC (heart). Notably, case #35 had no apparent clinical signs prior to death, and its prominent pathological changes were necrotizing pancreatitis, interstitial pneumonitis, necrotizing splenitis, nonsuppurative encephalitis, and myocardial degeneration, essentially consistent with acute toxoplasmosis [24]. Given the high virulence of the isolate, we hypothesized that immune dysregulation in this monkey (potentially associated with aging or pancreatitis) may have triggered the reactivation of *T. gondii*, leading to the progression from chronic to acute infection. However, the absence of antemortem clinical examinations made it difficult to confirm the primary cause of death for case #35.

MAT has been used extensively to detect IgG antibodies against *T. gondii* in humans and animals, and its validity for NHPs was supported by the diagnosis of toxoplasmosis or isolation of *T. gondii* [10,11,25,26]. However, antibody responses have varied with the species of NHPs. *Sapajus* spp. developed high titers as indicated by MAT, frequently higher than 1:500 (the highest recorded was 1:51,200), versus low titers below 1:100 or no antibodies in *Saimiri* spp. [6,27]. In this study, *T. gondii* infection was confirmed by MAT at titers of 1:1,600 in a black-capped capuchin (*Sapajus apella*), 1:32 in a black mangabey (*Lophocebus aterrimus*), and 1:8,192 in an orangutan (*Pongo pygmaeus*). Based on published data, the 1:8,192 titer of case #43 is the highest recorded for NHPs of the Hominoidea. The implications of differences in MAT titers for NHP species remain to be determined by further studies. Previous studies have reported a MAT titer of 1:16 in both a squirrel monkey that died of toxoplasmosis and a rhesus monkey from which *T. gondii* was isolated [10,25]. A MAT titer of 1:8 may indicate persistent infection with *T. gondii* in NHPs [10,11,28]. Here, a viable *T. gondii* strain was isolated from a black mangabey (*Lophocebus aterrimus*) with a titer of 1:32 in heart juice, further supporting the validity of MAT for NHPs.

Some monkeys have been seronegative for *T. gondii*, although they died of toxoplasmosis, especially squirrel monkeys (*Saimiri* spp.) [7,24,25,29,30]. This may have been due to the delayed humoral immune response to *T. gondii* infection (IgM for 1 week, IgG for 2 weeks). In such cases, PCR and IHC may be more sensitive than serological techniques, especially for the diagnosis of dead NHPs suspected of acute toxoplasmosis. In all samples, the lung, spleen, and myocardium were the most frequent positive tissues detected by PCR and IHC (S1 Table), suggesting that these three organs had a higher *T. gondii* parasite load in NHP cases. Thus, lungs, spleen, and myocardium are recommended as priority organs for the clinical detection of toxoplasmosis in NHPs. Other researchers have suggested that the lungs and liver should be selected for rapid diagnosis of toxoplasmosis by PCR or IHC [24,25]. Moreover, the high positivity rate in the lungs suggests that aerosol or droplet transmission may contribute to the horizontal spread of *T. gondii* in the captive NHP colony, consistent with the speculation in previous studies [30,31].

The TgMonkeyCHn3 strain was genotyped as ToxoDB #6 through PCR-RFLP analysis. To our knowledge, this is the first time that ToxoDB #6 has been identified in NHPs. This genotype belongs to the Type I variant, Clade A, and it was initially isolated (strain FOU) from a kidney transplant recipient (the infection origin was unknown) in 1992 [32,33]. ToxoDB #6 (*Africa* 1, Br1) is an epidemic genotype in Africa and Brazil, and it has been identified and reported in other regions [34]. The global distribution of ToxoDB #6 isolates from animals is summarized in Fig 2 and S4 Table. Prior to the present study, there were no reports of this genotype existing in East Asia, and thus, its origin remains unknown.

The geographic distribution of black mangabeys is restricted to tropical Africa near the equator, specifically the lowland rainforest areas of the southwest Congo Basin reaching to northern Angola [35], where the *Africa* 1 genotype is dominant. In recent years, many wild animals, including NHPs, have been introduced into Chinese zoos from tropical Africa; however, detection for *T. gondii* was often neglected in animal quarantine, and *T. gondii* oocysts could have attached to the transport vehicles and survived for a long time. It was speculated that the source of TgMonkeyCHn3 was most

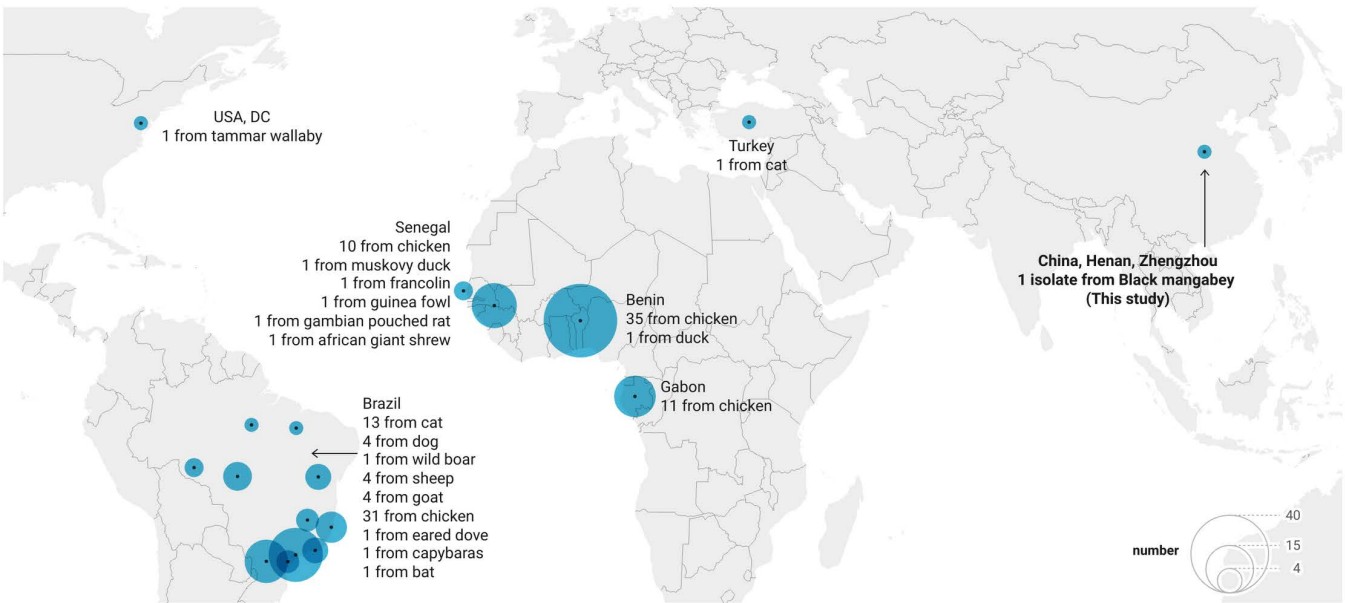

**Fig 2. Global distribution of ToxoDB genotype #6 strains from animals (n = 125).** Detailed information of these ToxoDB #6 strains is presented in S4 Table (base layer of the map: https://www.datawrapper.de/_/0fZ3d/?v=5).

likely attributed to the introduction of zoo animals from Africa, meaning that the black mangabey (case #35) was already infected with this strain in its original habitat. Anatolia is bridge between Africa, Europe and Asia. Finding viable *Africa* 1 genotype *T. gondii* isolates from newborns and cat from Turkey also supported the hypothesis of probable transcontinental transmission [36,37].

According to our investigations at the zoo of case #35, many feral cats and birds could move around different enclosures and come in contact with other animals and their feed, another possible source of transmission. Furthermore, previous studies have demonstrated a correlation between heavy precipitation and toxoplasmosis outbreaks in animals [38]. If there were felines infected with the ToxoDB #6 strain, the heavy rainstorm flooding in Zhengzhou on July 20, 2021 [39] could also have led to transmission through oocyst contamination in the water. It remains unknown whether other animals or humans are infected with TgMonkeyCHn3. In addition to the aerosol transmission in the NHP colonies as previously discussed [30,31], sexual transmission via semen [40] and endogenous transplacental infection [41] may have contributed to the dissemination of TgMonkeyCHn3. Fortunately, the female mate (case #40) of case #35 was not infected with *T. gondii*. In conclusion, it is necessary to conduct a complete investigation of *T. gondii* infection in animals and staff at the local zoo.

Two highly polymorphic rhoptry proteins (ROP18 and ROP5) have been demonstrated to be the main virulence determinants in *T. gondii* [22]. According to the previous mortality evaluation in mice, 33 *T. gondii* isolates with ROP18/ROP5 allele type of 1/3 indicated that this combination predicted a high virulence (average 90% mortality) [22,42]. Here, the cumulative mortality of TgMonkeyCHn3 infected mice was 71.4% (15/21, CI: 49.79%–86.44%), which was lower than predicted. The $LD_{100}$ was $10^2$ tachyzoites, and all mice infected with ≥$10^2$ tachyzoites died of acute toxoplasmosis at 11.4 ± 3.7 DPI, suggesting that TgMonkeyCHn3 is virulent in mice [43]. Overall, the ToxoDB #6 strains are generally virulent for mice worldwide (S4 Table). The survival times of 39.1% (27/69, CI: 28.47%–50.94%) ToxoDB #6 isolates were not significantly different from those of TgMonkeyCHn3, whereas the other 60.9% (42/69, CI: 49.06%–71.53%) isolates showed shorter survival times than TgMonkeyCHn3 (*P* < 0.05). Moreover, a case study in France reported severe human toxoplasmosis,

with molecular evidence of ToxoDB #6 in immunocompetent patients imported from West or Central Africa (including the distribution area of black mangabey) [44]. Therefore, more attention should be paid to the dissemination and pathogenicity of ToxoDB #6 strains in humans and animals.

In this study, a viable *T. gondii* strain, TgMonkeyCHn3 (ToxoDB #6), was successfully isolated from a black mangabey. The emergence of this virulent strain has augmented the genetic diversity of *T. gondii* in China, thereby presenting an escalating challenge for preventing and controlling this pathogen. It is necessary to take measures to control the activity of feral cats and provide effective *T. gondii* vaccine immunization to intermediate hosts (NHPs, marsupials) in zoos. Captive felids may shed *T. gondii* oocysts and thereby threaten environmental safety. Thus, oocysts in feces should be inactivated by burning or high-temperature treatment. Most importantly, customs quarantine measures should include checking for *T. gondii* infection in imported animals.

The present study on the presence of *T. gondii* in captive NHPs has several limitations. First, there was a limited number of serum and fresh tissue samples for in-depth study. A second limitation is that our understanding of the specificity and sensitivity of the diagnosis (MAT, PCR, mice biology) of *T. gondii* infection in various species of NHPs remains limited. This study may provide a basis for research on *T. gondii* infection in NHPs and humans.

## Supporting information

**S1 Fig. Workflow diagram for this study.**
(TIF)

**S2 Fig. *Toxoplasma gondii* distribution in tissue samples from non-human primates by immunohistochemical staining. (A, B)** Numerous *T. gondii* tachyzoites (arrow) are enclosed in parasitophorous vacuoles, spleen, case #29. **(C)** Focally distributed *T. gondii* tachyzoites (arrow) in the spleen of case #30. **(D, E, F)** *T. gondii* tachyzoites gathered in the interstitium of the lung (arrow) in case #33. **(G)** *T. gondii* cysts in cardiac fibers with a nearly elliptical shape, case #35. **(H)** The shed cell in the alveolar cavity was filled with *T. gondii* tachyzoites, case #41. Primary antibody: polyclonal rabbit anti-*T. gondii* antibody. Secondary antibody: anti-rabbit IgG conjugated with HRP/DAB. Bar = 50 μm.
(TIF)

**S3 Fig. Morphology of *Toxoplasma gondii* TgMonkeyCHn3. (A)** Dozens of *T. gondii* cysts (arrow) in the brains of Swiss mice (Tox#20–53, M#997) 30 days post-inoculation, smear, unstained, bar = 50 μm. **(B)** A parasitophorous vacuole ruptured to release several tachyzoites (arrow), lung, Swiss mouse (Tox#20–54, M#7), 15 days post-inoculation, smear, unstained, bar = 50 μm. **(C)** Hundreds of *T. gondii* bradyzoites (triangle) enclosed in a thin cyst wall (arrow), Swiss mouse (Tox#20–55, M#21) brain, 112 days post-inoculation, smear, unstained, bar = 50 μm. **(D)** Several tachyzoites enclosed in a parasitophorous vacuole (Pv), cell cultures. The apical ring (Ap), conoid (Co), nucleus (Nu), nucleolus (No), rhoptries (Rh), micronemes (Mn), lipid body (Lb), dense granules (Dg), and amylopectin (Am) are visible. TEM, bar = 2 μm. **(E)** A vesicle in exocytosis, presumably containing tubulovesicular membranes (arrow). TEM, bar = 1 μm. **(F)** In the transverse section of tachyzoites, well-developed mitochondria (Mi), regularly arranged microtubules (Mt), and pellicles (Pe) are visible. The tubulovesicular membranes (Tm) are located around the tachyzoites. TEM, bar = 500 nm.
(TIF)

**S4 Fig. Genotyping of *Toxoplasma gondii* TgMonkeyCHn3 strain isolated from Black mangabey.** 1: GT1, 2: PTG, 3: CTG, 4: TgCgCal, 5: MAS, 6: TgCatBr5, 7: TgCatBr64, 8: TgToucan (TgRsCr1), *: TgMonkeyCHn3, and M: Marker.
(TIF)

**S5 Fig. Original picture of genotyping of the isolate TgMonkeyCHn3.** M: Makers; 1: GT1; 2: PTG; 3: CTG; 4: MAS; 5: TgCgCa1; 6: TgCatBr5; 7: TgCatBr64; 8: TgRsCr1; *: TgMonkeyCHn3.
(PPTX)

**S1 Table. Background and *Toxoplasma gondii* infection in non-human primates from zoos (2022–2023, n = 17).**
NA: not available; ND: not done; F: female; M: male; Fixed: formalin fixation; MAT: modified agglutination test; PCR: polymerase chain reaction; IHC: immunohistochemistry; Sp: spleen; L: lung; H: heart.
(DOCX)

**S2 Table. Genotyping of *Toxoplasma gondii* TgMonkeyCHn3 strain by PCR-RFLP.**
(DOCX)

**S3 Table. Virulence evaluation of *Toxoplasma gondii* TgMonkeyCHn3 strain in Swiss mice.** 10-fold serial dilution concentrations from $1 \times 10^4$ to <1 *T. gondii* tachyzoite/mL were prepared using sterile PBS. Each group of Swiss mice (n = 5) was inoculated by intraperitoneal injection with 1 mL of tachyzoites at different dilutions. *T. gondii* cysts were found only in one mouse's brain (n = 80, 23 DPI) from group $10^2$ tachyzoites. DPI: Days post infection.
(DOCX)

**S4 Table. Global distribution of *Toxoplasma gondii* ToxoDB genotype #6 strains from animals and comparison of survival time in mice with TgMonkeyCHn3 (1992–2025).** [a]**: Comparing survival times of infected mice between TgMonkeyCHn3 and *T. gondii* ToxoDB genotype #6 strains using the asymptotic log-rank test for pair groups.**[b]**: The mouse survival times in bioassay of the strain compared with overall survival times ($10^0$-$10^4$ tachyzoites) of TgMonkeyCHn3.**[c]**: The mouse survival times of the strain compared with TgMonkeyCHn3 at the same or very close dose of tachyzoites.** NA: Not available. ND: Not done due to the small sample size.
(DOCX)

**S5 Table. Raw data of isolation viable *Toxoplasma gondii* from monkey tissues by mice.**
(XLSX)

## Acknowledgments

We thank Niuping Zhu and Hongjie Ren (Henan Agricultural University, China) for performing sample collection and Chunlei Su (University of Tennessee, USA) for carefully checking the TgMonkeyCHn3 genotypes and virulence factors. We thank Caili Zhang and Xianghua Liu of the TEM Center of Henan University of Chinese Medicine for their help with the transmission electron microscopy experiments.

## Author contributions

**Conceptualization:** Yurong Yang.

**Data curation:** Yiheng Ma, Liulu Yang, Yurong Yang.

**Funding acquisition:** Yurong Yang.

**Investigation:** Yiheng Ma, Liulu Yang, Yurong Yang.

**Methodology:** Yurong Yang.

**Project administration:** Yurong Yang.

**Resources:** Liulu Yang, Yurong Yang.

**Supervision:** Yurong Yang.

**Validation:** Yurong Yang.

**Visualization:** Yurong Yang.

**Writing – original draft:** Yiheng Ma, Yurong Yang.

**Writing – review & editing:** Yurong Yang.

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
