## [Decision Letter · Decision Letter 0]

PNTD-D-25-00020

Direct evidence of Black mangabey (Lophocebus aterrimus) as intermediate host of Toxoplasma gondii through isolation viable strain

Dear Dr. Yang,

Thank you for submitting your manuscript to PLOS Neglected Tropical Diseases. After careful consideration, we feel that it has merit but does not fully meet PLOS Neglected Tropical Diseases's publication criteria as it currently stands. Therefore, we invite you to submit a revised version of the manuscript that addresses the points raised during the review process.

Please submit your revised manuscript within 60 days Apr 27 2025 11:59PM. If you will need more time than this to complete your revisions, please reply to this message or contact the journal office at plosntds@plos.org. Please include the following items when submitting your revised manuscript:

We look forward to receiving your revised manuscript.

Kind regards,

Mehmet Aykur

Guest Editor

Abhay Satoskar

Section Editor

Shaden Kamhawi

co-Editor-in-Chief

Paul Brindley

co-Editor-in-Chief

**Additional Editor Comments:**

Thank you very much for submitting your manuscript "Direct evidence of Black mangabey (Lophocebus aterrimus) as intermediate host of Toxoplasma gondii through isolation viable strain" for consideration at PLOS Neglected Tropical Diseases. As with all papers reviewed by the journal, your manuscript was reviewed by members of the editorial board and by several independent reviewers. In light of the reviews (below this email), we would like to invite the resubmission of a significantly-revised version that takes into account the reviewers' comments.

**Editorial comment;**

The authors report the occurrence of Toxoplasmosis infection in non-human primates in a zoo in China. It is reported that for the first time, live Toxoplasma was isolated from the tissues of a Black mangabey and detected by different diagnostic methods. In addition, the effects of the isolation of the obtained Toxoplasma gondii strain and virulence were investigated. Although the study presents important findings, it is suggested that the statistical analysis, methodology should be handled more clearly and understandably as a common criticism of all reviewers, and the reviewer criticisms regarding the interpretation of virulence data should be addressed. When these deficiencies are addressed, the study can make a significant contribution.

**Journal Requirements:**

1) Please provide an Author Summary. This should appear in your manuscript between the Abstract (if applicable) and the Introduction, and should be 150-200 words long. The aim should be to make your findings accessible to a wide audience that includes both scientists and non-scientists. Sample summaries can be found on our website under Submission Guidelines:

2) We notice that your supplementary Figures, and Tables are included in the manuscript file. Please remove them and upload them with the file type 'Supporting Information'. Please ensure that each Supporting Information file has a legend listed in the manuscript after the references list.

3) Some material included in your submission may be copyrighted. According to PLOSu2019s copyright policy, authors who use figures or other material (e.g., graphics, clipart, maps) from another author or copyright holder must demonstrate or obtain permission to publish this material under the Creative Commons Attribution 4.0 International (CC BY 4.0) License used by PLOS journals. Please closely review the details of PLOSu2019s copyright requirements here: PLOS Licenses and Copyright. If you need to request permissions from a copyright holder, you may use PLOS's Copyright Content Permission form.

Potential Copyright Issues:

- Figure 4. Please provide a direct link to the base layer of the map (i.e., the country or region border shape) and ensure this is also included in the figure legend; and provide a link to the terms of use / license information for the base layer image or shapefile. We cannot publish proprietary or copyrighted maps (e.g. Google Maps, Mapquest) and the terms of use for your map base layer must be compatible with our CC BY 4.0 license.

4) In the online submission form, you indicated that "The data that support the findings of this study are available on request from the corresponding author". All PLOS journals now require all data underlying the findings described in their manuscript to be freely available to other researchers, either

- In a public repository

- Within the manuscript itself

- Uploaded as supplementary information.

5) Please ensure that the funders and grant numbers match between the Financial Disclosure field and the Funding Information tab in your submission form. Note that the funders must be provided in the same order in both places as well.

**Reviewers' Comments:**

Reviewer's Responses to Questions

**Key Review Criteria Required for Acceptance?**

**Methods** :

-Are the objectives of the study clearly articulated with a clear testable hypothesis stated?

-Is the study design appropriate to address the stated objectives?

-Is the population clearly described and appropriate for the hypothesis being tested?

-Is the sample size sufficient to ensure adequate power to address the hypothesis being tested?

-Were correct statistical analysis used to support conclusions?

-Are there concerns about ethical or regulatory requirements being met?

Reviewer #1: ok

Reviewer #2: PCR conditions (cycling parameters, reaction mix, controls, primer sequences) are missing. The modified agglutination test (MAT) lacks specificity details, such as: were positive and negative controls included? How was MAT sensitivity validated for NHP samples?

The methodology states that Swiss mice and IFN-γ knockout mice were used, but does not explain why was IFN-γ knockout used? What were the humane endpoints for euthanasia? The statistical rationale for the number of mice used (n=5 per group) is not justified.

The IHC methodology lacks specificity details, such as what antibody dilution and staining conditions were used?

TEM images are provided, but sample preparation details (fixation method, contrast staining) are missing.

Also, PCR conditions should included in the text.

Reviewer #3: 1) Page 4, Lines 8-9: please give reference for giving seropositivity to 1:8 dilution

2) page 4, Lines 11-14: Which tisues were used, how were the tissue collected (In terms of how the cross contamination was prevented)? How was the digestion performed? please describe it. Please give details abput DNA extraction? PCR method has to be described in detail.

3) Page 4, Line 19: what is the source of antibody?

4) Page 5, line 1: did you give tissues directly or digested it before use? please give the protocol.

how did you inoculate the tissues?

5) Page 5, line 4: Examined by which method?

6) page 5, line 5: did you previously check the mouse for T. gondii Ab? Please explain the details of the test to detect T. gondii Ab.

7) page 5, line 7: please describe the CV-1 cell culture and inoculation method In detail

8) page 5, line 9: please give detailed information about multiplex PCR

9) page 5, line 10-11: Please give details of the method about typing and please give detailed information about the reference strains

10) page 5, line 12-14: please give detailed information about this method

11) please add Kaplan-Meier survival curve to your statistical section

Reviewer #4: In this study, serological, molecular, and bioassay methods were used to investigate T.

5 gondii infection in 17 captived NHP in zoos from China between 2022 and 2023.

In the study, it was stated that the live strain of Toxoplasma gondii was determined for the first time in Black mangabey and that this animal was defined as a new host for T. gondii. Molecular characterization of T.gondi was performed.

However, it was observed that a wide variety of methods were used. Generally, the reference is given without giving the details of these methods. Page 10, Result Line 6 should review the expression "31 December 31". It would be useful to show the workflow in the Method section with a picture.

Reviewer #5: (No Response)

**Results** :

-Does the analysis presented match the analysis plan?

-Are the results clearly and completely presented?

-Are the figures (Tables, Images) of sufficient quality for clarity?

Reviewer #1: ok

Reviewer #2: The study reports infection rates of 83.3% in New World monkeys vs. 16.7% in Old World monkeys, but: No statistical test (e.g., chi-square test) is used to determine if the difference is significant. Confidence intervals (CIs) are not provided for prevalence rates.

Also, hte survival curve analysis (Figure 3) lacks statistical validation (e.g., Kaplan-Meier analysis, log-rank test). The study states that TgMonkeyCHn3 was highly virulent, but does not compare survival outcomes with previously reported ToxoDB #6 strains.

Reviewer #3: 1) Page 6, lines 6-13: please make a simple group name for mouse other than giving complex names such as Tox# 20-55 etc.

2) Please give the details of genotyping, for example show did you reach ToxoDB#6 aand gene allele type of ROP18/ROP5 was 3/3.

3) please change figure 3 explanation as “Kaplan-Meier survival curve of Swiss mice infected with various doses of T. gondii TgMonkeyCHn3 strain tachyzoites

4) Please cite the data showed on figure 4 in a separate table.

Reviewer #4: The analysis results are adequately expressed in tables and figures.

Reviewer #5: (No Response)

**Conclusions** :

-Are the conclusions supported by the data presented?

-Are the limitations of analysis clearly described?

-Do the authors discuss how these data can be helpful to advance our understanding of the topic under study?

-Is public health relevance addressed?

Reviewer #1: ok

Reviewer #2: Discussion

One major limitation is the missing comparision between other Toxo strains and the one found in this study. How does its virulence compare to other ToxoDB #6 strains isolated in Africa/Brazil? Does its ROP18/ROP5 (3/3) allele type match other virulent strains? Those questions needs to be adressed.

What is the possible transmission route in the Zoo? Needs more discussion. Any contamination on food? Or water? Or etc.???

What kind of preventions should take in to consideration

Figures & Tables:

Figure 1 (IHC images) should include scale bars and antibody information.

Figure 3 (Survival curves) needs log-rank test results.

Grammar needs revision by a native speaker

Reviewer #3: page 8, lines 15-16: Some monkeys were T. gondii seronegative, although they died of toxoplasmosis [13]. IHC and PCR may be more sensitive than serological techniques.” You have to discuss this results clearly and support your comment with references.

Reviewer #4: The authors did not report any limitations.

The conclusions are supported by the data presented

public health relevance İS addressed?

Reviewer #5: (No Response)

**Editorial and Data Presentation Modifications?**

Reviewer #1: none

Reviewer #2: (No Response)

Reviewer #3: (No Response)

Reviewer #4: (No Response)

Reviewer #5: (No Response)

**Summary and General Comments** :

Reviewer #1: The authors report toxoplasmosis in non- human in captive primates in China. They report isolation of viable Toxoplasma from tissues of a Black mangabey for the first time from this host; which is of scientific interest. This is detailed investigation using proper methods. Results are worthy of publication, however, the paper is too long.

Suggestions for shortening the paper are:

1.Title-First isolation of viable Toxoplasma gondii from a –

2.page 4, line 4-replace analysis with evaluation

3. page 4 line 13 replace checked with examined—also in rest of the paper

Page 5, line 8, replace into with on to

Move Table 1 to Supplemental Table

Describe Case 35 in the text

Page 7-lines 15-19-replace with a short sentence-findings were confirmed ultrastructurally

Figures 1 and 2 move to supplemental Figures

Supplemental Table 1-This Table is confusing-Column 1 state if it is based on counted tachyzoites or 10-fold dilutions-remove % ages-it is obvious

C olumn 2-No. infected/ No. inoculated

Column 3—remove DPI

Delete last column-state in foot note

Reduce discussion-focus on new findings

Reviewer #2: The article submitted by Ma et al. İs in the field of journal and investigating the T. gondii infection in Non-human Primates from Zoo. The authors did serological, molecular and bioassay-based evidence confirming that Black Mangabey (Lophocebus aterrimus) can serve as an intermediate host for T. gondii, and they successfully isolate a viable strain (TgMonkeyCHn3). The study highlights the strain’s high virulence and its ToxoDB genotype #6, previously reported in Africa but not in China.

While the study presents important findings, there are major concerns regarding novelty justification, statistical analysis, methodology transparency, and interpretation of virulence data that need to be addressed before publication.

Introduction

The introduction briefly mentions that New World monkeys are more susceptible than Old World monkeys, but does not explore why susceptibility differs (e.g., immune system variations, parasite strain differences). If it is something to do with the parasite strain, there should be more reference about this issue.

Authors states that this is the first report of genotype #6 in China, but does not compare findings with previous T. gondii genotyping studies in China or NHP studies worldwide. If there is any article mentioned on genotype#6, especially in China?

Reviewer #3: This is an interesting T. gondii isolation study from non-humans primates. There are some issues that should be clarified. My comments are attached.

The Title “Direct evidence of Black mangabey (Lophocebus aterrimus) as intermediate host of Toxoplasma gondii through isolation viable strain” should be changed as

“Demonstration of Black mangabey (Lophocebus aterrimus) as an intermediate host of Toxoplasma gondii through live strain isolation”

Reviewer #4: 

This study is important in terms of the transmission dynamics of T. gondii. It also contains important results in terms of public health, as it emphasizes that there may be new hosts in natural life and zoos and that they may differ in terms of virulence.

However, further detailing the method section and explaining the workflow with a picture would improve the quality of the article.

Reviewer #5: The authors conducted a very comprehensive study to demonstrate the presence of T. gondii in NHM and to isolate and genotype T. gondii. According to the results, the presence of T. gondii in NHM was detected by different diagnostic methods and ToxoDB genotype #6 was identified. I think the manuscript can be accepted for publication, but major revision are needed.

The recommendations are as follows.

1. The introduction is very weak and should be extended. T. gondii genotypes detected in NHM in previous studies should be mentioned.

2. Please check the statement “75.0-98.5% genetic homology”.

3. Tissue homogenization for bioassay should be specified in more detail.

4. The genotyping methodology should be specified in more detail.

5. What is ToxoDB genotype #6? Is it genotype I, II, III or atypical genotype? Please provide more detailed information.

6. It is not clear what Tox# 20-53 or Tox# 20-54 are. Please correct such statements.

PLOS authors have the option to publish the peer review history of their article (what does this mean? ). If published, this will include your full peer review and any attached files.

**Do you want your identity to be public for this peer review?** For information about this choice, including consent withdrawal, please see our Privacy Policy .

Reviewer #1: No

Reviewer #2: **Yes: ** Mehmet Karakus

Reviewer #3: No

Reviewer #4: No

Reviewer #5: No

**Figure resubmission:**
---

## [Decision Letter · Decision Letter 1]

PNTD-D-25-00020R1First Isolation of Viable Toxoplasma gondii from a Black Mangabey (Lophocebus aterrimus) Reveals the Emergence of the Africa 1 Lineage in East AsiaPLOS Neglected Tropical DiseasesDear Dr. Yang, Thank you for submitting your manuscript to PLOS Neglected Tropical Diseases. After careful consideration, we feel that it has merit but does not fully meet PLOS Neglected Tropical Diseases's publication criteria as it currently stands. Therefore, we invite you to submit a revised version of the manuscript that addresses the points raised during the review process. Please submit your revised manuscript within 30 days May 21 2025 11:59PM. If you will need more time than this to complete your revisions, please reply to this message or contact the journal office at plosntds@plos.org. Please include the following items when submitting your revised manuscript: * A rebuttal letter that responds to each point raised by the editor and reviewer(s). You should upload this letter as a separate file labeled 'Response to Reviewers '. This file does not need to include responses to any formatting updates and technical items listed in the 'Journal Requirements' section below. * A marked-up copy of your manuscript that highlights changes made to the original version. You should upload this as a separate file labeled 'Revised Manuscript with Track Changes '. * An unmarked version of your revised paper without tracked changes. You should upload this as a separate file labeled 'Manuscript '. If you would like to make changes to your financial disclosure, competing interests statement, or data availability statement, please make these updates within the submission form at the time of resubmission. Guidelines for resubmitting your figure files are available below the reviewer comments at the end of this letter. We look forward to receiving your revised manuscript. Kind regards, Mehmet AykurGuest EditorPLOS Neglected Tropical Diseases Abhay SatoskarSection EditorPLOS Neglected Tropical Diseases

Shaden Kamhawi

co-Editor-in-Chief

Paul Brindley

co-Editor-in-Chief

**Journal Requirements:**

Please ensure that the funders and grant numbers match between the Financial Disclosure field and the Funding Information tab in your submission form. Note that the funders must be provided in the same order in both places as well:

2) State what role the funders took in the study. If the funders had no role in your study, please state: "The funders had no role in study design, data collection and analysis, decision to publish, or preparation of the manuscript.".

**Reviewers' comments:** Reviewer's Responses to Questions

**Key Review Criteria Required for Acceptance?**

**Methods:**

-Are the objectives of the study clearly articulated with a clear testable hypothesis stated?

-Is the study design appropriate to address the stated objectives?

-Is the population clearly described and appropriate for the hypothesis being tested?

-Is the sample size sufficient to ensure adequate power to address the hypothesis being tested?

-Were correct statistical analysis used to support conclusions?

-Are there concerns about ethical or regulatory requirements being met?

Reviewer #1: yes

Reviewer #2: Yes

Yes

Yes

Yes

Yes

Yes

Reviewer #3: OK

Reviewer #4: It has been determined that the authors have made corrections and additions to each section of the article in line with the reviewers' suggestions, and the deficiencies of the article have been corrected and improved. Especially the method section has been explained in detail. The workflow diagram has been added.

Reviewer #5: (No Response)

**Results:**

-Does the analysis presented match the analysis plan?

-Are the results clearly and completely presented?

-Are the figures (Tables, Images) of sufficient quality for clarity?

Reviewer #1: yes

Reviewer #2: (No Response)

Reviewer #3: OK

Reviewer #4: It has been determined that the authors have made corrections and additions to each section of the article in line with the reviewers' suggestions, and the deficiencies of the article have been corrected and improved. Especially the method section has been explained in detail. The workflow diagram has been added.

Reviewer #5: (No Response)

**Conclusions:**

-Are the conclusions supported by the data presented?

-Are the limitations of analysis clearly described?

-Do the authors discuss how these data can be helpful to advance our understanding of the topic under study?

-Is public health relevance addressed?

Reviewer #1: yes

Reviewer #2: (No Response)

Reviewer #3: OK

Reviewer #4: It has been determined that the authors have made corrections and additions to each section of the article in line with the reviewers' suggestions, and the deficiencies of the article have been corrected and improved. Especially the method section has been explained in detail. The workflow diagram has been added.

Reviewer #5: (No Response)

**Editorial and Data Presentation Modifications?**

Reviewer #1: 1. Remove all duplication

2. Figures 1 and 2 move to supplementary section

3 Condense mouse mortality discussion

Swiss Webster outbred mice

Reviewer #2: (No Response)

Reviewer #3: OK

Reviewer #4: It has been determined that the authors have made corrections and additions to each section of the article in line with the reviewers' suggestions, and the deficiencies of the article have been corrected and improved. Especially the method section has been explained in detail. The workflow diagram has been added.

It can be published.

Reviewer #5: (No Response)

**Summary and General Comments:**

Reviewer #1: (No Response)

Reviewer #2: (No Response)

Reviewer #3: Authors have mentioned that “We added "Emergence of Africa 1 lineage" because it explicitly connects the isolated strain to its potential epidemiological origin (from Africa), which aligns with the host's native range, highlighting the likely transcontinental transmission through wildlife trade and the public health challenge of this virulent isolate. We believe that this new title reflects the main findings and highlights of the article.” Also they have changed their title based on this outcome.

There are publications where Africa 1 strain have been isolated from humans and stray cat which is outside Africa. You have not discussed these results in your discussion section. Please discuss these isolation and you may want to cite some of them also.

1) Can H, Döşkaya M, Ajzenberg D, Özdemir HG, Caner A, İz SG, Döşkaya AD, Atalay E, Çetinkaya Ç, Ürgen S, Karaçalı S, Ün C, Dardé ML, Gürüz Y. Genetic characterization of Toxoplasma gondii isolates and toxoplasmosis seroprevalence in stray cats of İzmir, Turkey. PLoS One. 2014 Aug 15;9(8):e104930. doi: 10.1371/journal.pone.0104930. PMID: 25127360; PMCID: PMC4134241.

2) Döşkaya M, Caner A, Ajzenberg D, Değirmenci A, Dardé ML, Can H, Erdoğan DD, Korkmaz M, Uner A, Güngör C, Altıntaş K, Gürüz Y. Isolation of Toxoplasma gondii strains similar to Africa 1 genotype in Turkey. Parasitol Int. 2013 Oct;62(5):471-4. doi: 10.1016/j.parint.2013.06.008. Epub 2013 Jun 28. PMID: 23811201.

Reviewer #4: Dear Authors, thank you for your effort to revise this manuscript.

It has been determined that the authors have made corrections and additions to each section of the article in line with the reviewers' suggestions, and the deficiencies of the article have been corrected and improved. Especially the method section has been explained in detail. The workflow diagram has been added.

It can be published.

Kind regards

Reviewer #5: The authors have addressed all the suggestions, except for providing a more detailed description of the genotyping methodology.

PLOS authors have the option to publish the peer review history of their article (what does this mean? ). If published, this will include your full peer review and any attached files.

**Do you want your identity to be public for this peer review?** For information about this choice, including consent withdrawal, please see our Privacy Policy .

Reviewer #1: No

Reviewer #2: **Yes: ** Mehmet Karakus

Reviewer #3: No

Reviewer #4: No

Reviewer #5: No

**Figure resubmission:** While revising your submission, please upload your figure files to the Preflight Analysis and Conversion Engine (PACE) digital diagnostic tool, https://pacev2.apexcovantage.com/. PACE helps ensure that figures meet PLOS requirements. To use PACE, you must first register as a user. Registration is free. Then, login and navigate to the UPLOAD tab, where you will find detailed instructions on how to use the tool. If you encounter any issues or have any questions when using PACE, please email PLOS at figures@plos.org. Please note that Supporting Information files do not need this step. If there are other versions of figure files still present in your submission file inventory at resubmission, please replace them with the PACE-processed versions.**Reproducibility:** To enhance the reproducibility of your results, we recommend that authors of applicable studies deposit laboratory protocols in protocols.io, where a protocol can be assigned its own identifier (DOI) such that it can be cited independently in the future. Additionally, PLOS ONE offers an option to publish peer-reviewed clinical study protocols. Read more information on sharing protocols at https://plos.org/protocols?utm_medium=editorial-email&utm_source=authorletters&utm_campaign=protocols

---

## [Editor Report · Decision Letter 2]

Dear Dr Yang,

We are pleased to inform you that your manuscript 'First Isolation of Viable Toxoplasma gondii from a Black Mangabey (Lophocebus aterrimus) Reveals the Emergence of the Africa 1 Lineage in East Asia' has been provisionally accepted for publication in PLOS Neglected Tropical Diseases.

Best regards,

Mehmet Aykur

Guest Editor

Abhay Satoskar

Section Editor

Shaden Kamhawi

co-Editor-in-Chief

Paul Brindley

co-Editor-in-Chief

---

## [Editor Report · Acceptance letter]

Dear Dr Yang,

We are delighted to inform you that your manuscript, " First Isolation of Viable Toxoplasma gondii from a Black Mangabey (Lophocebus aterrimus) Reveals the Emergence of the Africa 1 Lineage in East Asia ," has been formally accepted for publication in PLOS Neglected Tropical Diseases.

Best regards,

Shaden Kamhawi

co-Editor-in-Chief

Paul Brindley

co-Editor-in-Chief
